# FindZebra online search delving into rare disease case reports using natural language processing

Valentin Liévin[1,2], Jonas Meinertz Hansen[2], Allan Lund[3], Deborah Elstein[4], Mads Emil Matthiesen[2], Kaisa Elomaa[5], Kaja Zarakowska[6¤], Iris Himmelhan[6], Jaco Botha[6], Hanne Borgeskov[7], Ole Winther [1,2,8,9] *

1 DTU Compute, Technical University of Denmark, Lyngby, Denmark, 2 FindZebra, Denmark, 3 Centre Inherited Metabolic Diseases, Department of Clinical Genetics and Paediatrics, Copenhagen University Hospital, Rigshospitalet, Copenhagen Ø, Denmark, 4 Independent consultant; Jerusalem, Israel, 5 Takeda Oy, Helsinki, Finland, 6 Takeda Pharmaceuticals International AG, Zürich, Switzerland, 7 Department of Clinical Pharmacology, Aalborg University Hospital, Aalborg, Denmark, 8 Department of Biology, University of Copenhagen, Copenhagen N, Denmark, 9 Genomic Medicine, Copenhagen University Hospital, Rigshospitalet, Copenhagen Ø, Denmark

¤ Current address: UCB Farchim SA, Bulle, Switzerland
* ole.winther@bio.ku.dk

**Data Availability Statement:** This project relies on two data sources: a collection of PubMed abstracts (https://huggingface.co/datasets/findzebra/case-reports) and clinical Takeda owned Outcome

## Abstract

Early diagnosis is crucial for well-being and life quality of the rare disease patient. Access to the most complete knowledge about diseases through intelligent user interfaces can play an important role in supporting the physician reaching the correct diagnosis. Case reports may offer information about heterogeneous phenotypes which often further complicate rare disease diagnosis. The rare disease search engine FindZebra.com is extended to also access case report abstracts extracted from PubMed for several diseases. A search index for each disease is built in Apache Solr adding age, sex and clinical features extracted using text segmentation to enhance the specificity of search. Clinical experts performed retrospective validation of the search engine, utilising real-world Outcomes Survey data on Gaucher and Fabry patients. Medical experts evaluated the search results as being clinically relevant for the Fabry patients and less clinically relevant for the Gaucher patients. The shortcomings for Gaucher patients mainly reflect a mismatch between the current understanding and treatment of the disease and how it is reported in PubMed, notably in the older case reports. In response to this observation, a filter for the publication date was added in the final version of the tool available from deep.findzebra.com/<disease> with <disease> = gaucher, fabry, hae (Hereditary angioedema).

## Author summary

Rare diseases affect a substantial part of the population. However, they are especially challenging to diagnose. Because of their rarity, physicians often ignore rare diseases in the differential diagnosis. When confronted with hard-to-diagnose patients, physicians often

Survey data for Fabry (FOS) and Gaucher (GOS). De-identified records from 4484 Fabry and 1095 Gaucher patients served as the basis for selecting 20 Fabry and 20 Gaucher patients with atypical symptoms used in the expert validation. These 40 patient cases are available in S1 Data. Inquiries about the Takeda Outcome Survey can be addressed to GMA.Research@Takeda.com.

**Funding:** The study was funded by Takeda. OW and VL are supported by the Novo Nordisk Foundation (NNF20OC0062606) and DeepMind/ Google through their employment at UCph and DTU.

**Competing interests:** I have read the journal's policy and the authors of this manuscript have the following competing interests: KE, IH and JB are employed by Takeda and hold Takeda stocks/stock options. HB is a former employee in Takeda Pharma A/S, Denmark, holding a current position at Department of Clinical Pharmacology, Aalborg University Hospital in Denmark. KZ was employed by Takeda at the time of the study and holds Takeda stocks/stock options. OW, MM, VL and JH are employed by FindZebra, which received funding from Takeda for conducting the study. AL received reimbursement from FindZebra for clinical expertise in this study. AL reports also personal consultancy fees and travel grants from Takeda during the study, as well as grant support paid to his institution. DE received consultancy fees from Takeda for clinical expertise in this study.

turn to online resources like Google or PubMed, which index both general disease information as well as case reports. Case reports are a unique asset in helping the diagnosis of rare diseases because they often present with a varied and complex phenotype, which might not appear in the general literature. Nonetheless, searching for patient-relevant case reports is challenging. A tool dedicated to searching case reports assisting diagnosis is still missing because general-purpose search engines, like PubMed search, are primarily set up for literature search and because advanced search tools like FindZebra do not handle case reports.

In this study, we present a novel online search tool https://deep.findzebra.com/ dedicated to searching PubMed case reports based on a patient description (age, sex, symptoms, negative findings, etc.). Two medical experts evaluated the tool on forty challenging cases (twenty Fabry and twenty Gaucher). To our knowledge, this is the first specialized search tool for case reports that is built to assist diagnosis. This study provides a clear recipe for building and validating modern medical information retrieval systems to index and search complex and heterogeneous data.

## Introduction

A disease is considered rare when it affects: in Europe 1:2000 and in the US about 1:1600 people. Currently, there are more than 6000 distinct rare diseases in the EU [1]. Around 80% of rare diseases are of genetic origin and, of those, 70% manifest already in childhood. Many rare diseases are chronic, progressive, and life-threatening. Early diagnosis may save lives, slow disease progression and/or prevent further irreversible organ damage, and ultimately improve the quality of life for these patients. A recent population-based telephone survey in Germany revealed a median duration of the diagnostic delay of 20 or more years for some rare lysosomal storage disorders (LSD) [2].

The diagnostic odyssey is complicated since the signs and symptoms may intuitively implicate more common pathologies, and most primary care professionals may have no experience with any of these disorders. Moreover, recognizing the disease trajectory is also complicated by variable ages of onset, the progressive natural history with change in manifestations with age, variable presentation of multiple and diverse organs and tissues, and the lack of awareness of specialised diagnostic markers. In today's era of quick access to the internet and social media it is becoming equally true that patients find their diagnosis acting upon irrelevant information garnered from trawling these sources [3,4]. Two challenging rare diseases with complex phenotypes, a sufficient amount of patients available and good expert understanding are Fabry and Gaucher.

Fabry disease is a rare X-linked inherited LSD [5]. Major morbidity of several organ systems often begins in childhood with gastrointestinal, dermatological, and often ocular signs; as patients age, cardiac and renal disease may rather rapidly culminate in end-stage organ failure; stroke and other cardiovascular events that are life-threatening are also common. Early intervention with disease-specific therapies, enzyme replacement therapies (ERT), or pharmacological chaperones (PC) is critical and universally recommended before the development of more devastating and irreversible renal, cardiac, and/or cerebrovascular signs [6].

Gaucher disease has a clinical spectrum from a perinatal lethal neuronopathic form (type 2), to a form that has variable neurological and visceral signs (types 3a, 3b, and 3c), to a chronic, non-neuronopathic form (type 1), where some patients are truly asymptomatic

through-out their normative life-spans but others suffer from mild to severe visceral and hae-matological signs that appear variously anywhere from childhood to old age [7]. All these patients can benefit from early administration of several disease-specific treatment options to some extent: ERT and SRT will improve visceral and haematological signs as well as well-being; the SRTs and PCs can partly impact the neurological trajectory in the neuronopathic forms as well as the encroaching signs of Parkinson disease and other Lewy Body Dementia symptoms in type 1 patients [8,9].

Given the diagnostic delay, there are only a few published algorithms to assist in earlier diagnosis for either disease and none of them have been deployed into clinical use [10–13].

PubMed is the most complete information source on medical scientific knowledge. It comes with good information retrieval capabilities but is not designed for aiding diagnosis. This is demonstrated when benchmarking on medical cases against dedicated rare disease tools like FindZebra [14,15]. FindZebra, a search engine that has been popular within the med-ical community for the last past ten years, indexes a collection of curated disease articles from sources such as OrphaNet, OMIM and Wikipedia. The articles mostly describe generic disease phenotypes and therefore FindZebra only offers limited coverage of the more exceptional dis-ease phenotypes, which are described in the specialised medical literature. A tool that applies advanced information retrieval techniques to searching the specialised literature has to our knowledge so far been missing.

The case reports registered on PubMed are great candidates for extending the coverage of FindZebra. Over two million case reports are registered, and each one of them covers a unique medical case. Case reports could be used to improve the clinical management of today's patients, for instance by tailoring the treatment to the patient profile, or by supporting the healthcare providers in their disease management choices [16,17]. However, case reports show higher variability in style and quality than the curated articles gathered from OrphaNet, OMIM, Wikipedia, etc. Searching case reports is thus a more complex task and care must be taken to prioritize the information that is essential to recognize disease phenotypes in both the case reports and the search queries (age, sex, symptoms, genetics, negative findings, etc.). The proposed tool relies on two main components (Fig 1) built on publicly available tools (Pub-MedBERT [18] and Solr [19]):

1. a deep learning model that allows transforming unstructured text documents into struc-tured patient profiles

2. a full-text search engine (Solr) that allows searching for similar patients across the profile dimensions.

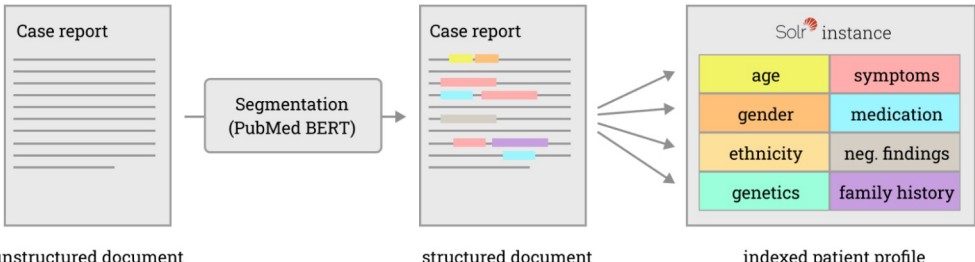

**Fig 1. Converting unstructured PubMed abstract into a structured search index using 1) text segmentation and 2) a Solr instance with composite fields corresponding to the text segmentation categories.**

In this study, we evaluate the tool based on real Fabry and Gaucher patients to identify issues particular to the task that will be essential for rolling the tool out to all rare diseases. In summary, the main contributions of the study are: 1) a recipe for building and validating an information retrieval system for heterogeneous medical information and 2) the search tool made available at deep.findzebra.com.

## Material and methods

We discuss how the novel search tool integrates with the existing FindZebra.com. We present the collection of case report abstracts indexed by our system before detailing the clinical data used to evaluate the tool. We conclude by describing the development of the search engine (segmentation of the abstracts and search ranking algorithm setup) and last the setup of the expert validation.

### FindZebra workflow

FindZebra.com allows searching across a collection of curated medical articles. For canonical disease profiles, this step is sufficient to find information that is relevant to the patient. For rare phenotypes, this new tool allows "diving in" the large pool of case reports within a particular disease to retrieve case reports that match the patient profile (Fig 2).

### Data

**PubMed case reports.** We collected case reports from 803 PubMed articles for Fabry disease and 883 for Gaucher disease. For each article, we retained only the abstract, which in most cases summarises information about the case at hand. We detail the data collection process in S1 Text.

**Clinical data—Fabry and Gaucher Outcomes Surveys.** We based the study on the real-world data from long-term observational Fabry and Gaucher Outcomes Surveys, FOS and GOS respectively. FOS and GOS aim at improving the clinical management of patients (see the S1 Text for further details). This was a non-interventional study, limited to the use of readily available data. It did not involve recontacting patients, and the informed consents, captured for the original FOS and GOS, allowed the use of their data for the validation purposes of this study. Data was anonymized by removing all information that could potentially identify a patient; a new randomization number was assigned to the Patient ID, all other potential

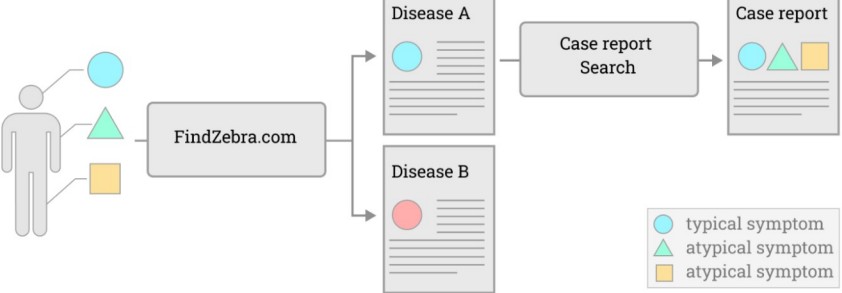

**Fig 2. Workflow for retrieving documentation relevant to atypical patients.** The patient presents with two typical findings where one leads to identification of Disease A. The case report search (the contribution of this paper) leads to a case report with the same atypical symptom combination.

**Table 1. Examples of survey data for Fabry and Gaucher patients.**

| Fabry patient F | | Gaucher patient G | |
|---|---|---|---|
| Diagnosis | Fabry | Diagnosis | Gaucher |
| Age | 66 | Age | 43 |
| Sex | male | Sex | male |
| CKD stage | 4 | Haemoglobin | 154 g/l |
| eGFR | 76·02 ml/min/1·73m$^2$ | Platelet count | 87 $10^9$/l |
| LVMI | 109·51 g/m**2·7 | Liver size | 4·60 multiples of normal |
| .. | .. | Spleen size | 0·85 multiples of normal |
| Symptoms | sign angiokeratomas, sign haemorrhoids, sign lv hypertrophy, symptoms vertigo, sign arrhythmia, haematuria, sign stroke, tumours, heart failure | Symptoms | lipid profile-low ldl, jaw-big osteolytic lesion, elevated ast, no hepatosplenomegaly |

identifiers, such as country, site name and date of birth were removed, as well as all other dates, e.g. visit dates and dates of laboratory assessments.

Records from 4484 Fabry patients and 1095 Gaucher patients were collected. Each record features demographic information (age, sex and mutation when available), a list of signs and symptoms and a quantitative evaluation of the relevant organs (Fabry: eGFR, LVMI; Gaucher: liver size, spleen size, haemoglobin value and platelet count).

We selected two anonymized patients, called patient F for Fabry and patient G for Gaucher. Their records are presented in Table 1. Throughout the text, we use patients F and G to showcase the data processing and evaluation steps.

**Converting survey entries to search queries.** We converted the survey data (tabular format) into full-text queries and numerical features into the corresponding signs using reference tables [20,21]. For instance, a low platelet count value was translated as "thrombocytopenia" whereas a normal value is converted as "no thrombocytopenia". The resulting textual features are combined into a comma-separated list of terms, see examples in Table 2.

We designed a segmentation model that transforms raw text into structured representations by extracting non-overlapping spans of text. Each span—or segment—is labelled using eight categories, which we summarise in Table 3. Each category is selected to represent a particular clinical feature that might be useful for diagnosis.

## Text segmentation

The model builds upon a domain-specific masked language model [22], PubMedBERT [18], which follows the same architecture as the popular BERT model [23]. BERT allows computing contextual language representations, which we augmented with a conditional random field likelihood to improve the local coherence of the segments [24].

**Table 2. Example of generated queries.**

| | Query |
|---|---|
| Patient F | male, elderly, 66-year-old, sign angiokeratomas, sign haemorrhoids, sign lv hypertrophy, sign arrhythmia, sign stroke, symptoms vertigo, haematuria, tumours, heart failure, severe chronic kidney disease |
| Patient G | 43-year-old, male, adult, thrombocytopenia, lipid profile-low ldl, jaw-big osteolytic lesion, elevated ast, no splenomegaly, no hepatosplenomegaly, normal haemoglobin level, no hepatomegaly |

**Table 3. Segmentation categories.** Each category represents a dimension of the patient profile. The categories are used to index the case reports and parse the queries.

| | Category | Examples |
|---|---|---|
| 1 | Sex | male, female, her, his |
| 2 | Age | young adult, 23-year-old, infant |
| 3 | Ethnicity | Ashkenazi Jewish, African American |
| 4 | Diagnosis, Signs and Symptoms | Fabry disease, Gaucher disease, Morbus Fabry, Chronic renal failure, recurrent posterior stroke-like symptoms, mild retardation, Gaucher cells, zebra bodies found in kidney biopsy, Creatinine level (200 micromol/L), anaemia, abnormal blood count |
| 5 | Medications and interventions | Enzyme replacement therapy, splenectomy |
| 6 | Genetics | c.427G>A (p.A143T) variant, rare mutation in the GBA gene |
| 7 | Negative findings | Covid-19 negative, no history of diabetes, normal blood count |
| 8 | Family history | History of early strokes in the family |

We randomly selected and labelled 100 abstracts for each disease. Each of the 200 abstracts were manually labeled into text segments. We chose to label spans of text such that each span of text encapsulates a single clinical feature completely. This results in segments of text that might overlap multiple text entities (see examples in Table 3). 20 documents were set aside for testing and the remaining documents were used for training (160) and validation (20). We detail the fine-tuning process in S1 Text. Our implementation relies on popular machine learning libraries [25–27].

We used the same model for both parsing the user queries and indexing the abstracts. To make the model robust to both the abstracts and the comma-separated queries, we augmented the training data by swapping abstracts with pseudo-queries for half the training iterations. Pseudo-queries were obtained by concatenating $N \sim \text{Poisson}(\lambda = 5)$ segments extracted from the replaced abstract.

## Search engine

The search engine is built on a composite Solr index for each disease separately. The ranking function is a weighted combination of the BM25 scores computed across each segmentation category. We detail the configuration and design of the index in S1 Text.

We analysed the corpora based on the extracted profiles. We used the SciSpaCy library to link symptoms to the Unified Medical Language System (UMLS) entities [28,29]. We report the frequency of symptoms as well as a summary of the demographic data in S1 Text.

## Validation protocol

The case report search engine is specifically designed for the use cases where the diagnosis is established or suspected but a deeper understanding of the phenotype is needed. Therefore, we focused on the subset of patients with atypical symptoms, which we defined in this study as the symptoms occurring in less than 10% of the population.

**Step 1—Preliminary non-expert validation.** During development, we inspected the quality of retrieval based on simple tests. For a selection of patients, we tested if the retrieved articles corresponded to the age, sex, mutations (if any) and the domain of symptoms (e.g., skeletal, psychiatric involvement, . . .).

**Step 2—Expert validation.** Two rare disease experts (DE and AL) evaluated the relevance of the search results given for the 20 patients for each disease. The anonymized data for the 40

patients is available in S1 Data. We selected patients with atypical phenotypes and diverse disease profiles. The experts were asked to evaluate the clinical relevance of each of the top three returned articles using a scale from one to five and using a text field. For each retrieved document, we report the maximum grade given by the two experts. For each patient, we report the precision for the top three results (P@3) based on the maximum grade and for multiple relevance thresholds. In S1 Text, we provide further details about the patient selection, the rating scale, the evaluation interface and experts' agreement.

**Step 3—Population and corpus level analysis.** To gain a better understanding of the diseases, their differences and how it affects retrieval, we analysed how the search engine maps the population of patients to the corpus of PubMed articles. We built a bipartite graph, using patients and articles as nodes, and created edges if an article was retrieved as top three. We used the resulting network to study the relationship between patients and PubMed articles.

### Ethics statement

This was a retrospective, non-interventional study, limited to the use of readily available patient data in Takeda-owned Fabry and Gaucher Outcomes surveys (FOS and GOS, respectively). The written informed consents obtained from the patients participating in the original FOS and GOS, allowed the use of their data for the validation purposes of this study which did not involve recontacting patients. Therefore, this study was not a subject for Ethics Committee approval. Data was furthermore anonymized by removing all information that could potentially identify a patient; a new randomization number was assigned to the Patient ID, all other potential identifiers, such as country, site name and date of birth were removed, as well as all other dates, e.g. visit dates and dates of laboratory assessments.

## Results

This section begins with a quantitative and qualitative evaluation of the final segmentation model. It continues with a description of the population of the selected patients. We then review the case report search tool in three acts: i) we display the case of two patients, ii) we present the expert review and iii) we illustrate how the search tools maps the cohort of patients to the PubMed corpus. As presented in S1 Text, the two diseases exhibit different profiles of symptoms, which supports the need to adapt the ranking function to each disease.

Text segmentation The final model scored 0·75 F1 score on the test set (0·76 F1 score on the validation set). The model appeared to be robust to a wide diversity of case reports and user queries. In Fig 3, we present a Fabry case report segmented using the final model. In Table 4, we present an example of a segmented query. In S1 Text, we present three additional labelled abstracts: one for Fabry, one for Gaucher and one out-of-domain example (COVID-19, see Supplement III).

### Clinical data and queries

We summarise the demographic features as well as the distribution of symptoms in S1 Text. We found that the populations of patients from the survey and from the PubMed articles follow similar demographics. Furthermore, we found that symptoms stated in the records were often discussed in the PubMed corpora.

The retrieval workflow (Fig 2) has two steps. For completeness, we also evaluate step 1 (FindZebra search) for all patients in the two Surveys. The correct diagnosis appeared in the top ten search results for 68·4% of the Fabry patients, whereas this was the case for only 21·7% of the Gaucher patients (see S1 Text for further details).

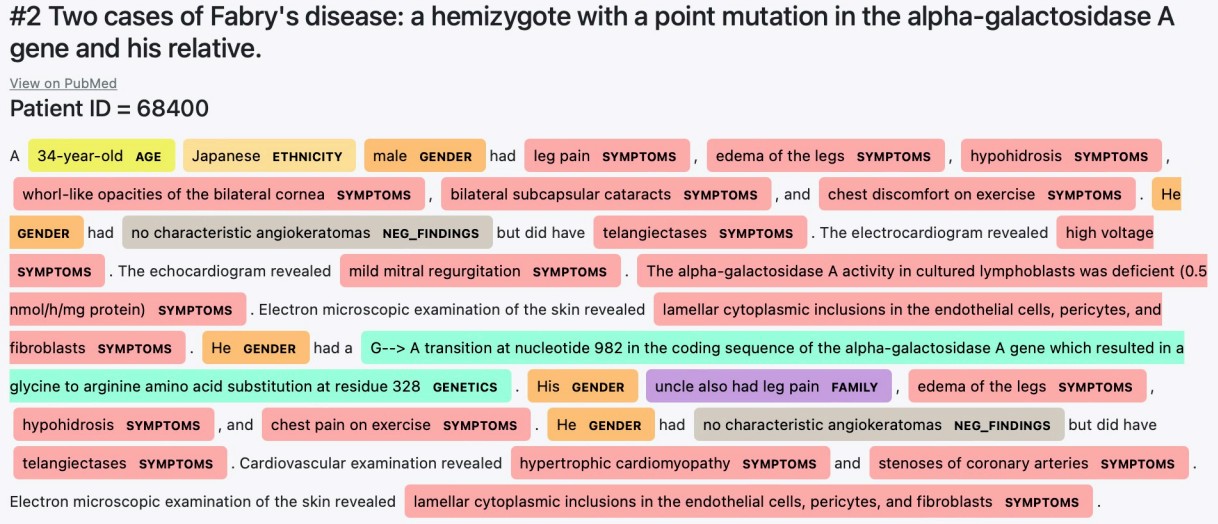

**Fig 3. Segmentation example of the article (test set): "Two cases of Fabry's disease: A hemizygote with a point mutation in the alpha-galactosidase A gene and his relative [30]".** Each colour corresponds to one of eight segmentation categories listed in Table 3.

## Subsets of typical and atypical patients

The medical expert validation of rare disease search (step 2 in Fig 2) focuses on the group of patients with atypical symptoms. Using the criteria defined in the previous section, we labelled 56% of the Fabry patients and 64% of the Gaucher patients as atypical. In Table 5, we report the number of patients for each group, the proportion of patients treated for Fabry or Gaucher and the mean number of symptoms recorded for each patient. We found that atypical patients have on average twice the number of symptoms and were more likely to be treated than the typical ones, indicating a more serious form of the disease.

## Expert validation

We found considerable disparities in the evaluation of the two diseases. Whereas retrieval was judged to be effective when applied to the Fabry patients, articles were more often judged as irrelevant for the Gaucher patients. We report the precision in Table 6 and display the distribution of maximum grades given to each article in Fig 4.

In the case of the Fabry patients, a minority of articles were judged irrelevant (11·7% of patients were assigned with a maximum rating lower than three) and 51·7% of the search results were graded with a maximum rating of at least four. The Gaucher patients were more

**Table 4. Example of segmented query.**

|  | Fabry patient F | Gaucher patient G |
|---|---|---|
| **age** | 66 | 43 |
| **sex** | male | male |
| **symptoms** | sign angiokeratomas, sign haemorrhoids, sign lv hypertrophy, sign arrhythmia, sign stroke, symptoms vertigo, haematuria, tumours, heart failure, severe chronic kidney disease | lipid profile-low ldl, jaw-big osteolytic lesion, elevated ast |
| **Negative findings** | - | no splenomegaly, no hepatosplenomegaly, normal haemoglobin level, no hepatomegaly |

**Table 5. Statistics for the groups of typical and atypical patients.**

| | Fabry | | | Gaucher | | |
|---|---|---|---|---|---|---|
| **Group** | **all** | **atypical** | **typical** | **all** | **atypical** | **typical** |
| **Patients** | 4484 | 2491 | 1993 | 1079 | 685 | 394 |
| **Treated (%)** | 56 | 66 | 39 | 82 | 84 | 79 |
| **Mean number of symptoms** | 6·6 | 9·8 | 3·1 | 5·6 | 6·6 | 3·6 |

difficult to match with relevant case reports, as 65·0% of the articles were rated one. Only 8·3% of the articles received at least one grade above three.

The experts' comments revealed six failure patterns listed in Table 7. The most common cause of failure (2 for Fabry, 18 for Gaucher) was attributed to retrieving articles that presented a radically different clinical picture, despite sharing a few symptoms and/or demographic features with the referenced patient. The second most prevalent cause of failure was associated with returning abstracts that were no longer considered valid by the medical experts. Other identified causes were diagnosis mismatch (failure pattern #3), missing data about the reference patient (#4), symptom mismatch (failure pattern #5), and age mismatch (failure pattern #6).

## Case study

In S1 Text, we illustrate the whole retrieval and validation process based on patients F and G. For each patient, we present the raw data, the segmented queries, and the retrieved abstracts associated with their corresponding expert ratings and comments.

## Population and corpora

To illustrate how the search scoring function maps a population of patients to the PubMed case reports, we sampled a subset of 500 patients for each disease to exclude the effect of the population size on the analysis. We created a patient-article network for each disease using the top three retrieved articles and the subset of patients. Both networks are visualised in Fig 5. We provide an analysis of the networks in S1 Text.

We recorded the number of retrieved articles for each disease as well as the mean number of patients connected to each article. We found significant differences between the two diseases: Fabry articles were connected to 7·6 patients on average (with a total of 295 retrieved articles) whereas Gaucher articles were connected to 3·9 patients per article (with a total of 386 articles).

**Table 6. Precision given 3 retrieved articles per patient (20 patients).** Each document is labelled as relevant if the maximum rating given by the two experts (AL and DE) is greater or equal to the threshold. The rating scale is described in S1 Text.

| Threshold | Fabry (P@3) | Gaucher (P@3) |
|---|---|---|
| Max. rating $\geq 2$ | 98·3% | 35·0% |
| Max. rating $\geq 3$ | 88·3% | 13·3% |
| Max. rating $\geq 4$ | 51·7% | 8·3% |
| Max. rating $\geq 5$ | 15·0% | 6·7% |

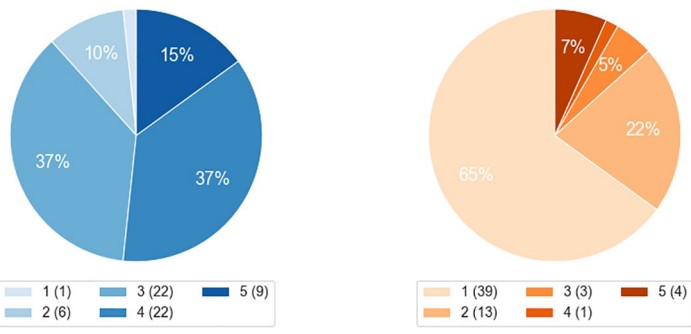

**Fig 4. Distribution of scores assigned to each document for each disease (Fabry disease left and Gaucher disease right).** For 20 patients per disease, we retrieve the top-3 abstracts. For each article, we use the maximum among the two expert ratings as evaluation score.

## Discussion

We have built a search engine that allows searching case reports based on patient features that are automatically extracted from the query and the indexed reports using deep learning. The tool allows case reports based on multiple features (sex, age, gender, mutation, symptoms, etc), which performs robustly thanks to the simple BM25-based design. Nonetheless, we tested more than semantic overall: we evaluated if the top three search results were clinically relevant according to medical experts. The articles were more often judged as clinically relevant for Fabry patients than for Gaucher patients, and we felt, looking at our results, that this could be partly explained by the dichotomy in explanations of the clinical manifestations and by the divergence in disease-specific management options of these two diseases.

Fabry disease is an X-linked disorder which implies that the male patients are generally more severely affected and at an earlier age than females plus there is also the impact of the various mutations that may be predictive of a specific phenotypic expression in both genders. On the other hand, Gaucher disease is usually divided into genotypes, and each genotype might lead to radically different disease trajectories (e.g., lethal, severe neuropathic genotype versus mild, non-neuropathic genotype). Therefore, matching Gaucher patients was more challenging, because the greater diversity of profiles made it easier to miss, and because age and sex were not as informative as in Fabry (Gaucher is not X-linked).

This highlights the limitations of our ranking function. In some cases, we found the ranking function to be misaligned with the expert judgement, as it placed too little weight on sex

**Table 7. Failure patterns.** Number of identified failure patterns for each disease.

| # | Pattern | # Fabry | # Gaucher |
|---|---|---|---|
| 1 | The article presents a rare or very specific clinical profile or treatment that is irrelevant to the patient despite significant lexical overlap and/or similar symptoms ("buzz words") | 2 | 18 |
| 2 | The article is outdated, its content is no longer clinically relevant | 2 | 11 |
| 3 | The case presented in the article presents similar clinical features, but the diagnosis was dissimilar (similar features, different causes) | 2 | 6 |
| 4 | The article studies a specific characteristic (mutation, ethnicity, family history), that is unknown for the reference patient | 3 | 3 |
| 5 | The case and the patients don't share similar symptoms, the case might have been matched solely based on the age and sex of the reference patient | 1 | 4 |
| 6 | Age mismatch (infant vs. adult) | 1 | 4 |

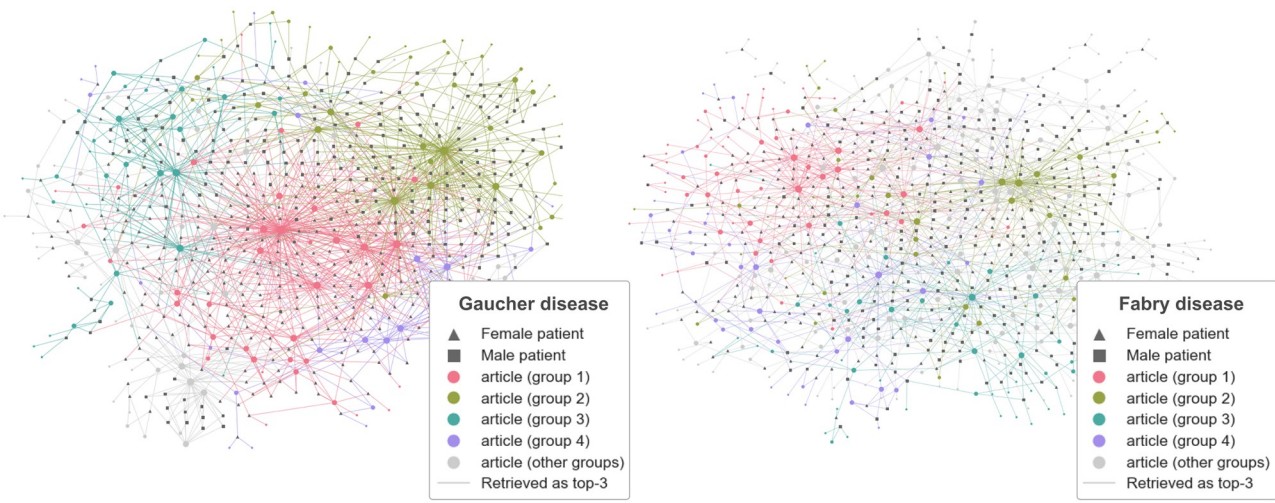

**Fig 5. Visualization of the patient-article networks for Gaucher (left) and Fabry (right).** Nodes represent patients and articles; edges are drawn if an article is retrieved as the top three for a given patient. Each colour corresponds to a cluster of patients and articles. This visualization shows how the population of patients maps to the corpus of articles. The Fabry network shows a higher degree of clustering than the Gaucher Network.

(failure pattern #6), or on symptoms (failure pattern #5). In other cases, failure was attributed to BM25, as it only allows handling symptoms independently, thus failing to grasp the whole clinical picture. This might lead to placing too much weight on a few rare terms ("buzz words"), which is linked to failure pattern #1.

Furthermore, we found that many retrieved articles were outdated, especially in the case of the Gaucher disease (failure pattern #1), this is explained by the recent changes in the clinical management of the two diseases. In Fabry disease, the triad of end-stage organ failure of renal, cardiac, and cerebrovascular events in patients may be partly prevented, but they are still irreversible once established. The ramifications on the other organ systems remain poorly controlled, which may also impact quality of life and longevity, so that patients today face many of the same challenges as those of decades past. However, disease management in Gaucher disease has been transformed. Several new modalities of therapeutics can now assure normative function by reversing visceral signs and symptoms in the non-neuropathic patients. Furthermore, the disease phenotypes have evolved due to a tendency to diagnose earlier and due to longer survival. Ultimately, this was seen in our study which underscored the explosion of recent developments for the several types of Gaucher, so that older case reports were of limited value for patients being seen today.

This uncovers a broader problem that not all case reports contain valid information. Whereas validating the content of case reports is challenging, recency can be easily controlled. For the released version of the search tool, we added the possibility of filtering on publication date.

The validation protocol was designed to mimic the real-world use of our tool, but a discrepancy remains between the evaluation setup and the real-world usage. First, we used all the recorded data for each patient, whereas in a clinical context, the healthcare professionals rely only on the subset of the features that might be relevant in that particular clinical setting. Second, the generated queries only contained information about age, sex, mutation, symptoms and negative findings. Our tool handles additional profile dimensions such as medications, family history and ethnicity (which is a critical factor in the management of Fabry and Gaucher diseases). Third, whereas information systems are traditionally evaluated using the top ten results, we restricted the evaluation to the top three results. Scrolling past the top three

results might be required in the more difficult cases such as the ones observed in the Gaucher evaluation. Lastly, we acknowledge the challenge and limitations of data anonymization especially in the rare disease space. However, as the FOS and GOS patient data collected for the purpose of this study is global and consists of relatively high numbers of patients, we consider anonymization sufficient.

## Conclusion

We have built a search engine specialised for case reports and submitted it to a thorough expert validation process using real-world clinical Outcomes Surveys data. To the best of our knowledge, our tool pioneers the task of indexing and retrieving cases reports with the aim of aiding diagnosis of rare diseases. It allows browsing large quantities of case reports natural language and clinical descriptions using a structured ranking function (age, sex, mutation, symptom, etc...). Our approach details a general approach that can be used to make the clinical literature more readily useful for healthcare practitioners.

Based on real-world rare diseases information, we found that retrieved articles were often clinically relevant for the Fabry patients, whereas articles retrieved for the Gaucher patients had less clinical value. Further analysis of the expert comments and the patient-abstract network allowed us to identify shortcomings associated with our method. Our study highlights the gap that remains between modern search technologies and clinical practice.

Although this study was restricted to the Fabry and Gaucher diseases, we will now focus on scaling the process to all rare diseases recorded at FindZebra.com. We learned from the expert evaluation and will use their feedback to improve our tool, by adding a temporal filter to the query field. It is hoped that our tool will be used in the field to help the healthcare professionals to improve the clinical management of the many patients who suffer from rare diseases. Our tool will remain publicly available.

## Supporting information

**S1 Text. The docx file contains the supplementary information referenced in the main text.**
(DOCX)

**S1 Data. This zip file contains the 40 patient cases in JSON format (20 Fabry and 20 Gaucher) used for the evaluation.**
(ZIP)

## Author Contributions

**Conceptualization:** Valentin Liévin, Mads Emil Matthiesen, Kaisa Elomaa, Ole Winther.

**Data curation:** Valentin Liévin, Jonas Meinertz Hansen, Jaco Botha, Ole Winther.

**Formal analysis:** Valentin Liévin, Jaco Botha, Ole Winther.

**Funding acquisition:** Mads Emil Matthiesen, Kaisa Elomaa, Iris Himmelhan, Hanne Borgeskov, Ole Winther.

**Investigation:** Valentin Liévin, Jonas Meinertz Hansen, Kaisa Elomaa, Iris Himmelhan, Ole Winther.

**Methodology:** Valentin Liévin, Jonas Meinertz Hansen, Mads Emil Matthiesen, Jaco Botha, Ole Winther.

**Project administration:** Mads Emil Matthiesen, Kaisa Elomaa, Kaja Zarakowska, Iris Himmelhan, Hanne Borgeskov, Ole Winther.

**Resources:** Valentin Liévin, Jonas Meinertz Hansen, Mads Emil Matthiesen, Kaisa Elomaa, Iris Himmelhan, Jaco Botha, Ole Winther.

**Software:** Valentin Liévin, Jonas Meinertz Hansen.

**Supervision:** Mads Emil Matthiesen, Kaisa Elomaa, Kaja Zarakowska, Hanne Borgeskov, Ole Winther.

**Validation:** Valentin Liévin, Allan Lund, Deborah Elstein.

**Visualization:** Valentin Liévin.

**Writing – original draft:** Valentin Liévin, Kaisa Elomaa, Ole Winther.

**Writing – review & editing:** Valentin Liévin, Jonas Meinertz Hansen, Allan Lund, Deborah Elstein, Mads Emil Matthiesen, Kaisa Elomaa, Kaja Zarakowska, Iris Himmelhan, Jaco Botha, Hanne Borgeskov, Ole Winther.

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
