## [Decision Letter · Decision Letter 0]

26 Dec 2022

PDIG-D-22-00278

FindZebra online search delving into rare disease case reports using natural language processing

PLOS Digital Health

Dear Dr. Winther,

Thank you for submitting your manuscript to PLOS Digital Health. After careful consideration, we feel that it has merit but does not fully meet PLOS Digital Health's publication criteria as it currently stands. Therefore, we invite you to submit a revised version of the manuscript that addresses the points raised during the review process.

EDITOR: 

We would like you to respond to the comments made by reviewers.

Please proof read your manuscript before submission. 

Please submit your revised manuscript within 60 days Feb 24 2023 11:59PM. If you will need more time than this to complete your revisions, please reply to this message or contact the journal office at digitalhealth@plos.org. Please include the following items when submitting your revised manuscript:

We look forward to receiving your revised manuscript.

Kind regards,

Ewen M. Harrison, PhD FRCS

Academic Editor

PLOS Digital Health

Journal Requirements:

2. We ask that a manuscript source file is provided at Revision. Please upload your manuscript file as a .doc, .docx, .rtf or .tex.

3. Please provide separate figure files in .tif or .eps format only and remove any figures embedded in your manuscript file. Please also ensure that all files are under our size limit of 10MB.

4. In the online submission form, you indicated that "This project relies on two data sources: a collection of PubMed abstracts and clinical Takeda owned Outcome Survey data for Fabry (FOS) and Gaucher (GOS). De-identified records from 4484 Fabry and 1095 Gaucher patients were used for validation purposes as described in the Material and Methods, and some de-identified raw data is released in the Results. Neither the entire raw Outcomes Survey data nor related documents will be made publicly available. The list of the PubMed articles used in this study will be made available on request.". All PLOS journals now require all data underlying the findings described in their manuscript to be freely available to other researchers, either 1. In a public repository, 2. Within the manuscript itself, or 3. Uploaded as supplementary information.

Additional Editor Comments (if provided):

Reviewers' comments:

Reviewer's Responses to Questions

**Comments to the Author**

1. Does this manuscript meet PLOS Digital Health’s publication criteria? Is the manuscript technically sound, and do the data support the conclusions? The manuscript must describe methodologically and ethically rigorous research with conclusions that are appropriately drawn based on the data presented.

Reviewer #1: Yes

Reviewer #2: Partly

2. Has the statistical analysis been performed appropriately and rigorously?

Reviewer #1: N/A

Reviewer #2: N/A

3. Have the authors made all data underlying the findings in their manuscript fully available (please refer to the Data Availability Statement at the start of the manuscript PDF file)?

Reviewer #1: Yes

Reviewer #2: Yes

4. Is the manuscript presented in an intelligible fashion and written in standard English?

Reviewer #1: Yes

Reviewer #2: Yes

5. Review Comments to the Author

Reviewer #1: Thank you for submitting this interesting work.

The authors describe the evaluation of extension of the search engine FindZebra with PubMed search for case reports based on patient characteristics. To do this, they conducted an extensive validation process by medical experts. The validation of the approach as performed against the Fabry and Gaucher disease. Feedback from the experts was highlighted and will be part of future research.

The study contributes to current research in NLP and rare diseases and is of high importance. The approach presented is a useful combination of established knowledge sources or useful extension of an established tool.

MAJOR ISSUES

(1) Focus of the article should be sharpened: Should the study describe the evaluation process or how the tool works, or both?

(2) Introduction:

(2a) The combined use of already existing models should be added when mentioning the two components for the tool (p. 3). Otherwise, the impression is created that the components were completely newly developed by the authors.

(2b) The goal of the study is stated as the evaluation of the tool (p. 3). However, the title and the methods also describe the functional area of the extension. Therefore, the research objective (and possibly also the title) should be modified accordingly.

(3) Material and methods:

(3a) When explaining the workflow, only the goal is named and not the method of functioning. A more detailed explanation – especially of Figure 2 – is necessary here (p. 3)

(4) Results:

(4a) The description for "Search index" should focus more on actual results and less on the procedure (p. 7).

(4b) It is not always clear which results were obtained by which methods; especially with respect to the validation process, it would be interesting to know which results resulted from which step. Here, a reference to the individual steps would be important. (An example of this is the evaluation of the previous FindZebra tool (p. 8)).

MINOR ISSUES

(5) Introduction:

(5a) Transition from explanations of diseases to explanation of PubMed could seem more natural by adding the research gap again (p. 2).

(5b) Claim that a tool is missing is not substantiated, so perhaps add "to our knowledge" (p. 3).

(5c) Incomplete sentence: “Case reports could be used to improve the clinical management of today’s patients, for instance by tailoring the treatment to the patient profile, or by supporting the healthcare providers in their [16,17]” (p. 3).

(5d) Figure 1 is referenced when the two components are mentioned, but the two components are not explicitly shown in the figure. Here, you could highlight the individual components separately (p. 3).

(6) Material and Methods:

(6a) The introduction to the chapter "Material and methods" does not exactly match the following subchapters. The title and sequences should be harmonized here (p. 3).

(6b) Is this really de facto anonymization? Or can individual patients be identified by the combination of sex, age and mutation (p. 4)?

(6c) Explanation of text segmentation should be under the heading "Text Segmentation" and not be content of "Data" (p. 5).

(7) Results:

(7a) The introduction to the chapter "Results" does not exactly match the following subchapters. The title and sequences should be harmonized here (p. 7).

(7b) Revision of figures and tables necessary:

(7b.i) Figure 4: Name / describe units (p. 10).

(7b.ii) Labeling of table 7 differs from other labeling (p. 9).

(7b.iii) No table 6 included (p. 8-9).

(7b.iv) Two figures are named “Figure 4” (p. 9 and 11).

(8) Conclusion:

(8a) Improve the conclusion by describing the novelty of the approach in more detail and by stating scientific and technical implications for the entire research field (p. 13).

OTHER POINTS

(9) Great unified and beautiful presentation of illustrations.

(10) Great detailed discussion of the results.

Reviewer #2: The manuscript describes a new search functionality for rare disease case reports of FindZebra.com service. Two rare diseases of Gaucher and Fabry were used as the exemplar study diseases. Natural language processing models/tools were utilised to support the modelling/indexing of case reports and the matching between user queries and case reports. Particularly, user queries were generated from 'real' patient cases including age/gender and clinical features like symptoms and phenotypes. Evaluation protocols and metrics were proposed to validate the utilities of the service in supporting clinical decision making for patients with those diseases, seemingly in scenarios of both diagnosis and treatments.

Overall, from clinical utility point of view, this would potentially be a very valuable work and much needed service for supporting rare disease diagnosis, treatment and managements. However, technically - from information retrieval and NLP point of view, the work requires further developments and clarifications to make it publishable - in other words, making substantial contribution to the field and useful for the community.

1. It is not clear how NLP models were used and developed. Named entity recognition was mentioned only in the abstract. In the main text and supplementary it was called segmentations. The two might be totally different NLP tasks. The use of terminology aside, there is no information how the NER was done. PubmedBERT was mentioned to be the language model for fine-tuning the segmentation task (assuming the NER for clinical features like symptoms etc). However, there was no mention where the ground truth of NER came from. 

2. The ranking algorithm is key in the methodology of an information retrieval system. But it seems very limited contribution was proposed to that aspect - the default BM25 was used instead.

3. The key evaluation result of precision@3 seems not very good. There were 5 grades as detailed in the supplementary. From the descriptions, only grade 4 & 5 can be assessed to be relevant. From table 7, the 'real' useful results for the two diseases were 52% and 8% for the two diseases respectively. This seems a bit low for a search service according to these numbers. There was no baseline provided to compare their service to. Also, there was no ablation evaluation on different techniques. Therefore, it is not clear how difficult the task was and it is hard to justify how good their service was.

4. From table 8 (analysis of the failure patterns), first, clearly the domain experts were evaluating the service from facilitating diagnosis/treatment points of view. For example, item 2 was "The article is outdated", which is clearly not a direct IR/NLP problem. However, the method/models proposed by the authors seem not dealing with these requirements directly. This leads to a general question - have the authors properly defined their technical tasks based on the intended use of the service?

5. A side and puzzled finding from table 8 was the big difference of the last two rows between the two diseases. There seems to be no obvious reasons why the second disease was harder for the same (relatively simpler) IR tasks.

6. I struggled to understand the relevance of "Population and Corpora" section. First, it seems not directly relevant to information retrieval and NLP tasks. Second, the networks/graphs were generated from top 3 articles of the service. However, the performance of the current service as it is seems not good enough to support the generation of a reasonably good network for this kind of analysis.

Presentation issues:

- There are no justifications on why the two rare diseases. Do they present two exemplar (distinct) IR/NLP challenges?

- The abstract does not provide clear descriptions on what the main validation purposes are.

- There are many abbreviations which should be expanded when they were first used.

- There are grammar and incomplete sentences at various places in the manuscript.

6. PLOS authors have the option to publish the peer review history of their article (what does this mean?). If published, this will include your full peer review and any attached files.

**Do you want your identity to be public for this peer review?** For information about this choice, including consent withdrawal, please see our Privacy Policy.

Reviewer #1: No

Reviewer #2: Yes: Honghan Wu

---

## [Editor Report · Decision Letter 1]

3 May 2023

FindZebra online search delving into rare disease case reports using natural language processing

PDIG-D-22-00278R1

Dear Prof. Winther,

We are pleased to inform you that your manuscript 'FindZebra online search delving into rare disease case reports using natural language processing' has been provisionally accepted for publication in PLOS Digital Health.

Best regards,

Sarah Mayo

Staff Admin

PLOS Digital Health